# CFTR Lifecycle Map—A Systems Medicine Model of CFTR Maturation to Predict Possible Active Compound Combinations

**DOI:** 10.3390/ijms22147590

**Published:** 2021-07-15

**Authors:** Liza Vinhoven, Frauke Stanke, Sylvia Hafkemeyer, Manuel Manfred Nietert

**Affiliations:** 1Department of Medical Bioinformatics, University Medical Center Göttingen, Goldschmidtstraße 1, 37077 Göttingen, Germany; liza.vinhoven@med.uni-goettingen.de; 2Clinic for Pediatric Pneumology, Allergology and Neonatology, Hannover Medical School, Carl-Neuberg-Strasse 1, 30625 Hannover, Germany; mekus.frauke@mh-hannover.de; 3Biomedical Research in Endstage and Obstructive Lung Disease Hannover (BREATH), The German Center for Lung Research, Carl-Neuberg-Strasse 1, 30625 Hannover, Germany; 4Mukoviszidose Institut gGmbH, In den Dauen 6, 53117 Bonn, Germany; shafkemeyer@muko.info; 5CIDAS Campus Institute Data Science, Goldschmidtstraße 1, 37077 Göttingen, Germany

**Keywords:** cystic fibrosis, CFTR, CFTR maturation, systems medicine model, trafficking, CFTR modulators

## Abstract

Different causative therapeutics for CF patients have been developed. There are still no mutation-specific therapeutics for some patients, especially those with rare CFTR mutations. For this purpose, high-throughput screens have been performed which result in various candidate compounds, with mostly unclear modes of action. In order to elucidate the mechanism of action for promising candidate substances and to be able to predict possible synergistic effects of substance combinations, we used a systems biology approach to create a model of the CFTR maturation pathway in cells in a standardized, human- and machine-readable format. It is composed of a core map, manually curated from small-scale experiments in human cells, and a coarse map including interactors identified in large-scale efforts. The manually curated core map includes 170 different molecular entities and 156 reactions from 221 publications. The coarse map encompasses 1384 unique proteins from four publications. The overlap between the two data sources amounts to 46 proteins. The CFTR Lifecycle Map can be used to support the identification of potential targets inside the cell and elucidate the mode of action for candidate substances. It thereby provides a backbone to structure available data as well as a tool to develop hypotheses regarding novel therapeutics.

## 1. Introduction

Cystic fibrosis (CF) is an inherited disorder prevalent among the white European population, where, with an incidence of approximately 1 in 3000 newborns, it is one of the most common monogenic autosomal recessive diseases [1]. CF is caused by mutations of the cystic fibrosis transmembrane conductance regulator (*CFTR*) gene [2], which encodes a membrane protein that serves as a chloride and bicarbonate channel in exocrine epithelia of various organs, and thereby regulates the viscosity of the mucus lining [3]. Defective CFTR, therefore, has severe implications throughout the body, its major hallmarks being recurrent pulmonary infections and pancreatic insufficiency [2,3].

CFTR is an approximately 170 kDa membrane-spanning glycoprotein composed of 1440 amino acid residues and complex glycosylation [4,5,6]. It belongs to the ATP binding cassette (ABC) Transporter Superfamily [7,8], but, unlike most of them, acts as an ion channel as opposed to an active transporter. As an ion channel, CFTR only requires ATP for opening, whereas other ABC transporters also require ATP for the active transport of substrates across membranes [9]. In accordance with the other ABC transporters, CFTR consists of two transmembrane domains (TM1 and TM2) and two cytosolic nucleotide-binding domains (NBD1 and NBD2) with an additional highly flexible regulatory region (R-region) at its center [4,9,10,11]. Several in-depth reviews cover the structure and opening mechanism of CFTR in great detail [9,12,13,14]. CFTR undergoes an intricate maturation pathway with complex folding and core glycosylation at the endoplasmic reticulum (ER), before being trafficked through the secretory pathway and further glycosylated at the Golgi apparatus. At the ER, CFTR is subject to extensive quality control mechanisms, often resulting in premature degradation through the ER-associated degradation pathway (ERAD), which leads to only 20–40% of nascent peptides of wild-type (wt) CFTR being correctly folded and trafficked to the apical plasma membrane (PM). After being integrated into the membrane, CFTR undergoes continuous endocytosis, recycling and, when misfolded, degradation in the lysosome [15].

To date, more than 2100 mutations of the *CFTR* gene are known, several hundred of which are known to be disease-causing [16,17,18]. They can cause defects anywhere in CFTR’s complicated and sensitive lifecycle, which is why they were traditionally subdivided into six different classes by the effect they have on the CFTR protein: (I) no protein synthesis, (II) CFTR trafficking defect, (III) dysregulation of CFTR, (IV) defective chloride conductance or channel gating, (V) reduced CFTR transcription and synthesis and (VI) less stable CFTR [19,20,21]. The traditional classification system has been proposed by Welsh and Smith in 1993 [19] and since has been reviewed and adapted in a range of publications [20,22,23]. However, since many mutations exhibit more complex phenotypes, nowadays, a modified and expanded classification system finds use, where all combinations of the six original mutation classes are regarded as possible classes [21]. For example, the most prevalent CF-causing mutation is the deletion of phenylalanine at position 508 (F508del), which accounts for almost 70% of CF chromosomes worldwide [1,24]. F508del causes a folding defect which results in premature degradation at the ER, making it a class II mutation [6,25]. However, even when rescued and trafficked to the membrane, its channel gating is reduced and it is less stable than wt-CFTR, meaning it exhibits mutation class III/IV and VI characteristics as well [26,27]. According to the expanded classification system, it is therefore classified as a class II-III-VI mutation [21]. Several other mutations also display different defects. Therefore, the wide range of *CFTR* gene mutations, resulting in different defects in the CFTR protein, makes causative treatments for CF difficult to find, and the recently available CF modulators do not target all mutations. As a result, there is still no mutation-specific therapeutic for some patients, especially those with rare *CFTR* mutations. The latest research efforts, therefore, focused on developing combination therapies to target multiple defects at once [28,29]. For this purpose, high-throughput (HT) screens have been performed [30,31,32,33,34,35,36,37,38,39,40,41,42], where thousands of substances have been tested in different cell models [43]. These result in a plethora of data and various candidate compounds, often with an unclear mode of action. In order to provide an overview of already tested compounds, we previously established the publicly available database CandActCFTR (https://candactcftr.ams.med.uni-goettingen.de/ (accessed on 6 June 2021)), where substances from 90 publications are listed and categorized according to their influence on CFTR function (manuscript submitted for publication).

In order to support the elucidation of the mechanism of action for promising candidate substances and to be able to predict possible synergistic effects of substance combinations, we used a systems biology approach to create a model of the CFTR maturation pathway in cells. Systems biology modeling aims to gather knowledge on biological systems and translate it into a human- and machine-understandable format in order to analyze its behavior and interactions. To make models reproducible and reusable, there are certain standards and formats to adhere to. The most well-established, standardized format in the graphical representation of biological processes is the Systems Biology Graphical Notation (SBGN), consisting of three languages [44]. Of the three, the SBGN Process Description (PD) language allows the most detailed representation of molecular mechanisms using nodes and directed edges. Molecular entities are shown as nodes and have different shapes depending on their molecular species. For example, proteins are represented as rounded rectangles, RNAs as parallelograms and small molecules as ovals. The reactions (e.g., transcription, translation and state transition) and reaction regulations (e.g., catalysis and inhibition) between the molecular entities are represented by edges, shaped as arrows with differently shaped heads depending on the type of interaction [44]. A glossary of systems biology and bioinformatics terms used here can be found in the Appendix A.

Here, we present a systems biological model of the CFTR lifecycle in a standardized, explorable and tractable format. The model is composed of two datasets, a core map manually curated from small-scale experiments in human cells, and a coarse map including interactors identified in high-throughput (HT) efforts [45,46,47,48]. Interactors are here defined as molecular entities that influence CFTR directly or indirectly. Both data layers are divided into submaps focusing on different stages of the CFTR lifecycle and different processes the ion channel is involved in. Overall, the manually curated core map includes 170 different molecular entities and 156 reactions from 221 experimental publications. The high-throughput data layer encompasses 1384 unique interactors from four publications by Wang et al., 2006, Pankow et al., 2015, Santos et al., 2019 and Matos et al., 2019 [45,46,47,48]. The CFTR Lifecycle map provides a tool to structure and exploit existing knowledge and data, as well as develop a hypothesis regarding synergistic drug targets and novel therapeutics.

## 2. Results

### 2.1. CFTR Map

The CFTR map encompasses information from small-scale experiments as well as high-throughput efforts, leading to differences in the degree of detail and confidence. It was therefore split into different data layers. The first data layer, the core map, was manually curated and only includes high-confidence interactors, confirmed by at least two independent small-scale experiments or acknowledged by two reviews from different research groups. As a result, the number of molecular interactions in the core map is limited but each is described with a high level of detail. The second data layer in the coarse map represents the high-throughput interactome of wt-CFTR (the interactome of F508del was excluded) as published by Wang et al., 2006, Pankow et al., 2015, Santos et al., 2019 and Matos et al., 2019 [45,46,47,48] in a structured cell layout. Here, the large-scale experimental method does not allow for conclusions regarding the nature of the interactions. Therefore, a small level of detail, but a high number of interactors, is included in the map. Overall, the manually curated core map comprises 170 different interactors and the coarse map from large-scale efforts contains 1384 interactors; 46 interactors could be found in both (Figure 1). A list of all interactors in both maps, as well as their overlap, can be found in Appendix A. To prevent redundancies, the overlapping interactors were only kept in the manually curated core map and excluded from the coarse map.

### 2.2. Representation of the CFTR Lifecycle in the CFTR Core Map

The CFTR core map represents the molecular mechanisms affecting wt-CFTR during its lifecycle. It is the product of an exhaustive literature curation process and the manual integration of different data sources. As the whole model is represented and written in the standardized SBGN format, it is human understandable as well as computationally tractable. It was created in the editor CellDesigner4.4.2 [49,50], adhering to the Process Description language of the SBGN format [44]. At the moment, it encompasses 262 different molecular entities and 156 reactions in 6 main cellular compartments. The biomolecules are categorized into 149 proteins, 58 complexes, 28 simple molecules, 13 ions, 6 genes, 5 RNAs, and 3 pools of degraded protein, amino acids, or nucleotides. Proteins can be subdivided into generic proteins, truncated proteins, ion channels, and receptor proteins. The color of the molecular entity indicates whether it was identified in at least one polarized cell line (green) or non-polarized cell lines only (yellow). Reactions are specified as state transitions, inhibitions, catalysis, transports, and heterodimer associations and dissociations. Each interaction is supported by at least two independent publications, leading to an overall number of 221 publications. A complete list of all interactors and references can be found in Appendix A. 

The CFTR core map (Figure 2) has a roughly cell-shaped layout, with CFTR making its way from the nucleus at the bottom all the way up to its site of action at the plasma membrane. It covers the molecular interactions CFTR undergoes on its way from being transcribed in the nucleus to being a functional ion channel at the apical plasma membrane, including its activity and regulation there, as well as endocytosis, recycling and degradation of the mature protein. The map can be subdivided into five submaps (Table 1) to enable the user to either look at its whole or individual processes, depending on their focus or interest. The five submaps are guided by the subcellular location and process they focus on.

Transcription—Nucleus: The Nucleus submap covers the transcriptional regulation of the *CFTR* gene into its mRNA.Translation, Folding and ER Quality Control—ER: The ER submap step-by-step describes the translation of the mRNA into the CFTR peptide and its integration into the membrane as well as folding steps modulated by chaperones, core glycosylation, and the calnexin cycle involved in ER quality control. Depending on the folding success, CFTR may progress through the secretory pathway or be degraded through ER-associated degradation.Secretory Pathway—ER, Golgi Apparatus, Plasma Membrane: The Secretory Pathway submap covers COPII vesicle-mediated trafficking between the ER, Golgi, and the plasma membrane and the full glycosylation at the Golgi apparatus. It also describes unconventional trafficking of the protein between the ER and plasma membrane, which has been found to be an alternative route CFTR may take.Activity and Regulation—Plasma Membrane: The Activity submap covers the phosphorylation-dependent activation of CFTR through the cAMP signaling cascade, channel opening, closing, and ion conductance as well as regulatory interactions with other ion channels and stabilization through interactions with the cytoskeleton.Endocytosis, Recycling and Degradation—Plasma Membrane, Endosomes, Lysosomes (Figure 2c): The final submap describes the endocytosis of the mature CFTR protein from the plasma membrane, which can be recycled back to the membrane or degraded in lysosomes.

As different cell lines can have different effects on the interactome of proteins [43,51], the cell lines used in the small-scale experiments were identified from each reference and categorized into polarized and non-polarized cells. As the question whether cells are to be characterized as polarized or non-polarized can often be a matter of debate when grown under experimental conditions, polarized cells are here defined as cells with the general ability to polarize. For each interactor, it is indicated by color whether or not they were identified in polarized (green color scheme) or non-polarized cell lines (yellow color scheme), whereby the specific cell lines for each interactor can be found in Appendix A. The percentage of interactors identified in at least one polarized cell line was calculated for each submap, amounting to 97% in the Transcription submap, 69% and 52% in the ER and Golgi submap, respectively, 82% in Activity and Regulation and 74% in Endocytosis, Recycling and Degradation.

### 2.3. Protein–Protein Interaction Network and Topological Analysis of the CFTR Core Map

To analyze the manually curated core map with regard to interactions between the proteins included, a protein–protein interaction network was created using the list of genes present in the model (Figure 3a). All proteins (nodes) of the protein–protein interaction network were identified as CFTR interactors through the manual literature curation, whereas all interactions (edges) between them were identified through the BioGrid database. A list of all interactions present in the protein–protein interaction network can be found in Appendix A. There are 145 nodes and 326 edges present in the network, and the average number of neighbors amounts to approximately 4.5. Another important property to assess when analyzing protein–protein interaction networks is its degree distribution. Here, the degree of a protein (node) is the number of interactions (edges) it shares with another protein. Most biological networks are considered ‘scale-free’, meaning that the majority of nodes have a low degree with only a few highly interconnected hubs, represented by a large diameter in Figure 3a. In order to analyze whether the CFTR protein–protein interaction network is scale-free, the degree distribution was calculated (Figure 3b). As can be seen, it follows a power law, confirming that it is, indeed, scale-free. Their scale-free character lends biological networks features important in biological systems. On the one hand, they are quite stable, as a random failure is likely to affect a protein with a low degree due to their high prevalence. Therefore, random failures and changes are unlikely to have a large effect on the overall network. On the other hand, targeted interventions at one of the few hubs will have a large effect on the whole network, which can be important when treating diseases and considering side effects. Here, CFTR is, as expected, the node with the highest degree, which can also be seen in the visualization in Figure 3a, where CFTR is the largest node. Apart from CFTR, the five nodes with the next highest degree are HSP90AA1 (heat shock protein 90), STUB1 (ubiquitin–protein ligase CHIP), UBE2I (SUMO-conjugating enzyme), NR3C1 (Glucorticoid receptor), and VCP (Transitional ER ATPase).

### 2.4. Visualization of the wt-CFTR Interactome as Coarse Model

In addition to the extensive manually curated interaction pathways, which include all high-confidence interactors and detailed interactions, a second data layer was included for interactors with lower confidence and detail. The second data layer represents information from large-scale experiments, namely the wt-CFTR core interactomes published by Wang et al., 2006, Pankow et al., 2015, Santos et al., 2019 and Matos et al., 2019 [45,46,47,48]. As the data are stored in long gene lists, in contrast to the detailed, text-based description of interactions in the manual model, different tools were used to represent the high-throughput interactome. The coarse map is also written in the SBGN format, but was created using libsbgnpy [52] and CellDesigner [49,50] and is represented in the SBGN Activity Flow notation, which lacks mechanistic information. The use of the python library libsbgnpy allowed for an automated construction of the maps. In order to structure the information into an intuitive, cell-based layout, the interactors were grouped according to their function and subcellular localization. Again, the model was divided into several submaps, based on the functional categorization (Table 2). The functional category of each interactor was solely based on information from the respective publications and not inferred from other sources. Interactors for which no function was indicated were assigned to the ‘other/unknown’ category. Figure 4 shows an exemplary image of one of the submaps, which is focused on Endocytosis, Recycling and Degradation. Each submap focuses on one main step or area of function in the CFTR-lifecycle, abstracted into a state-transition reaction. Six of them correspond to those from the core map, with an extra map for mRNA processing between the “Transcription” and “Translation, Folding and ER Quality control”. Additional maps focus on interactions with the cytoskeleton, as well as immune-related and other interactors.

Transcription—Nucleus: The Transcription submap focuses on the *CFTR* gene and the production of pre-mRNA. All interactors are divided into two functional categories, those that affect the gene directly, e.g., “DNA repair” and “replication”, and those that affect the transcription, such as “transcription” and chromatin structure”. Apart from the CFTR entities, it includes 17 nodes, seven affecting the gene and ten affecting the state transition.RNA processing: The additional RNA processing map describes the conversion of pre-mRNA to mature mRNA. It includes interactors with functional categorizations, such as nuclear export and RNA splicing, but also RNA degradation, and contains 36 nodes apart from CFTR.Translation, Folding and ER Quality Control—ER: The third submap summarizes the processes taking place in the ER in two state transitions. One is the processing from mature mRNA to folded, core-glycosylated CFTR peptide and degradation at any stage during ER quality control, resulting in an overall number of 45 interactors. The interactors are color-coded depending on whether they affect folding (57 interactors, green), degradation (one interactor, red), both (three interactors, red), or the interaction is unspecified (653 interactors, yellow). The 653 unspecified interactors are mainly from the data published by Santos et al. [47], where the authors characterize the interactome of CFTR prior to its exit from the ER.Secretory Pathway: In accordance with the core map, the Secretory Pathway submap shows the trafficking of the CFTR peptide between the ER, Golgi and PM after folding and core glycosylation. For reasons of simplicity and a lack of information, all 22 interactors were depicted as influencing CFTR trafficking between the ER and Golgi, even though they might be affecting different steps.Activity: Here, all reactions involved in the activity and regulation of and by mature CFTR within the plasma membrane PM are summarized as channel opening, influenced by 38 different entities. It also includes 145 unspecified interactors, that were reported to interact with CFTR at the PM [48], but for which the nature of the interaction is unclear.Recycling and Degradation (Figure 4b): This submap is split into the recycling and degradation of mature CFTR. Endocytosis-regulating interactors are included in the recycling category, resulting in 12 interactors affecting recycling and 32 influencing degradation.Cytoskeleton: An additional submap is designated for interactors with an influence on the anchoring of CFTR in the cell, including 62 entities apart from CFTR.Immunity: A separate submap shows nine interactors playing a role in immunity (10 interactors).Other Functions: In order to represent the whole datasets published, another submap includes all interactors that fall into none of the categories above. These include, for example, proteins involved in metabolism and those for which no function regarding CFTR could be specified (250 interactors).

### 2.5. Systemic Interpretation and Comparison of Manually Curated Model and Large-Scale Interactome

The lists of interactors from the core map and coarse map were compared for overlaps. Interestingly, of the 170 manually curated interactors and the 1384 interactors from the high-throughput screens, only 46 interactors could be found in both datasets (Figure 1).We next wanted to see whether the interactors in both datasets belong to similar functional categories, as this would indicate that similar pathways of importance for CFTR folding and maturation have been identified by the targeted analysis of interactors (core map) or by hypothesis-free high-throughput analysis (coarse map). 

Datasets of the core and the coarse map were analyzed using the BioInfoMiner web application [53], which performs a biological interpretation based on a list of genes, resulting in prioritized lists of systemic processes and genes, similar to a gene enrichment analysis. Here, the gene ontology (GO) [54,55] terms and Reactome Pathway Database [56] terms were assigned to all interactors. Reactome is a manually curated and peer-reviewed database for cellular pathways on a molecular level, whereas the gene ontology knowledgebase provides a model of biological systems to represent the current knowledge on the function of genes from the molecular to organism level. Therefore, while Reactome and Gene Ontology share a certain overlap, they mainly complement each other, as Reactome associates genes with specific molecular processes, whereas Gene Ontology also takes broader biological processes into account.

Of the top 20 prioritized Gene Ontology terms of the core model-derived gene list and of the coarse model-derived high-throughput gene list, 12 were shared between core and coarse model, including *cellular localization*, *macromolecule localization*, *protein localization* and *vesicle-mediated transport* (Figure 5). Furthermore, the heat shock protein *HSP90AA1*, as well as the ER ATPase *VCP* were found in the top 45 prioritized genes of both gene sets. 

In addition to the analysis using Gene Ontology terms, the BioInfoMiner analysis was repeated with terms from the Reactome Pathway Database [56]. Here, however, there was no overlap between Reactome terms assigned to the two gene lists derived from the core model and the coarse model, respectively. While the top-ranked terms for the manually curated gene list derived from Reactome were mostly intracellular transport-related, the top-ranked terms for the high-throughput gene list mostly related to the cellular response to stress and infection. The complete results of the analysis can be found in Appendix A. 

The diverging results between the Gene Ontology-based analysis and Reactome-based analysis most likely stem from the different levels of biological activity that are defined by these two databases. While the analyzed Gene Ontology terms mainly include the broad biological process terms, such as *intracellular transport*, the Reactome pathway terms are on a smaller, more detailed level, such as *SRP-dependent cotranslational protein targeting to membrane*. Consequently, the same major biological processes appear to be overrepresented in the manually curated core model and the high-throughput interactome represented in the coarse model, while different specific molecular pathways from Reactome are recognized in both layers of the CFTR Lifecycle Map.

## 3. Discussion

Extensive research has been dedicated towards the elucidation of the processes CFTR undergoes from its transcription on the way to becoming a folded, complex glycosylated and fully functional ion channel at the plasma membrane of apical epithelial cells. In the CFTR Lifecycle Map, we collected the knowledge accumulated by researchers over the course of three decades and represented it in a human-and machine-readable way. In doing so, we adhered to the standards established by the systems biology community. We applied common literature curation criteria and followed the well-established SBGN format, as well as MIRIAM guidelines to create and annotate a core map of the CFTR lifecycle. Additionally, we included data from large-scale efforts to identify the CFTR interactome in a coarse map as a second data layer.

When comparing the manually curated data from small-scale experiments within the core model with the data from high-throughput screens by Wang et al., 2006, Pankow et al., 2015, Santos et al., 2019 and Matos et al., 2019 [45,46,47,48] embedded into the coarse model, it becomes evident that the overlap between them is surprisingly small. To detect whether genes from the core and the coarse model belong to shared functional categories, BioInfoMiner was also used to prioritize genes in the core model and in the high-throughput generated interactome of the coarse model. BioInfoMiner is a tool used for the analysis of semantic networks and prioritizes key systemic processes and related genes present in a set of genes based on different databases [53]. Here, functionalities were assigned to the CFTR interaction partners based on Gene Ontology biological process terms [54,55] and the terms from the Reactome Pathway Database [56]. Gene Ontology provides information on the function of genes and gene products, which is derived from the scientific literature by the Gene Ontology Consortium. The Reactome Pathway Database is a manually curated and peer-reviewed database for molecular pathways by an international multidisciplinary team. The overlap between core model and coarse model genes detected by the Gene Ontology categories indicate that similar pathways of importance for CFTR folding and maturation have been identified by the targeted analysis of interactors (core map) or by hypothesis-free high-throughput analysis (coarse map). Using the Gene Ontology term analysis, two genes were prioritized among the top ten for both datasets. Both of them, HSP90AA1, better known as heat shock protein 90, and the ER ATPase VCP were also found amongst the hubs in the protein–protein interaction network of the manually curated interactors. Both VCP and HSP90AA1 are involved in a multitude of different cellular processes. It is therefore not surprising that they are highly connected in the protein–protein interaction network and are ranked as important genes in the Gene Ontology term analysis. They most likely also play an important role in CFTR maturation; however, they are very unspecific and likely influence the folding and maturation of proteins other than CFTR as well.

In contrast to the overlap in Gene Ontology categories, where entries of the core and the cores model were shared, the pathway terms from the Reactome Pathway Database differ substantially between the manually curated core model and the high-throughput-derived coarse model gene lists, which reinforces the poor overlap of only 46 interactors present in both layers of the CFTR Lifecycle Map. These differences on the small-scale level may stem from the different experimental approaches used to identify the interactors. While the manual curation from small-scale experiments includes a lot of information on the immature CFTR, the high-throughput screens use large-scale co-immunoprecipitation-based approaches to identify the interactome of the mature, or at least folded, wt-CFTR. These results, however, also indicate that, although a lot is known about the maturation and processing of CFTR, there are still substantial knowledge gaps. This is especially true for the interaction with other ion channels at the plasma membrane. In the last decade, other ion channels that may be in regulatory interaction with CFTR, such as the calcium-activated chloride channel Anoctamin-1 (also known as ANO1, TMEM16A and ORAOV2), or other chloride channels which are being discussed as CFTR alternatives, have been brought to the center of attention [57,58,59]. However, there is still work to be carried out in the elucidation of the CFTR regulatory network and its role in ion homeostasis at the plasma membrane, especially considering that potentially interacting ion channels, such as CLCA2 (also known as CACC3) and KCNJ1 (also known as ROMK) had to be rejected during the construction of the CFTR map due to a lack of experimental evidence. Knowledge gaps also seem to exist with regard to the intracellular trafficking pathway. It is known that CFTR is usually trafficked through conventional Golgi-mediated exocytosis, but may also circumvent the Golgi via an unconventional route [60,61,62]. This may be relevant for the rescue of misfolded variants such as F508del-CFTR [62,63,64,65]. Differences in the observed pathways and interactions may also arise from the variety of cell models used, which we addressed by color-coding interactors from polarized and non-polarized cells in our model. However, it becomes clear that the molecular mechanisms and interactors of both pathways are not fully elucidated yet, as can be seen in the lack of detail in the CFTR core map. 

The CFTR Lifecycle Map is part of the CandActCFTR project, which established a publicly available database of candidate cystic fibrosis therapeutics, combining data from different sources, such as high-throughput- and small scale screens, data from relevant databases and unpublished primary data (candactcftr.ams.med.uni-goettingen.de/). The CFTR Lifecycle Map, as a second part of the project, aims to provide the means to identify promising drug targets and elucidate the mode of action for candidate substances. Furthermore, it will be used to predict possible additive effects of different substance combinations. It thereby simultaneously provides a backbone to structure available data as well as a tool to develop hypotheses regarding novel therapeutics.

## 4. Materials and Methods

### 4.1. Creation of the Core Map

The map was drawn using the Process Description (PD) language of the Systems Biology Graphical Notation (SBGN) syntax [44] and the diagram editor CellDesigner4.4.2 [49,50]. The PD language of SBGN allows the representation of molecular biological models in a standardized manner using nodes and edges. The nodes, i.e., the molecular entities, of the network are represented by different symbols according to their molecular species. This map includes entities specified as genes, mRNA, proteins, truncated proteins, protein complexes, receptors, ion channels, simple molecules, and ions. The edges between molecular entities serve to specify reactions and reaction regulations. Reaction types present in the map include transcription, translation, state transition, complex formation and dissociation and transport, and reaction regulations are of the type catalysis, inhibition, physical stimulation, or modulation, where the exact nature of the interaction is unknown. Each molecular entity is named according to its HUGO-approved gene symbol [66] (https://www.genenames.org/ (accessed on 26 January 2021)) for genes and gene products, and ChEBI name [67] (www.ebi.ac.uk/chebi/ (accessed on 26 January 2021)) for simple molecules and ions. The model is split into six main cellular compartments (cytoplasm, plasma membrane, extracellular space, nucleus, endoplasmic reticulum (ER), and Golgi apparatus) and smaller compartments (vesicles and endosomes). Following the Minimal Information Required for the Annotation of Models (MIRIAM) Guidelines [68], an established standard for annotating systems biology models, all components are annotated by unique identifiers. The HUGO ID [66] (https://www.genenames.org/ (accessed on 26 June 2021)) is used for genes, and gene products, the UniProt ID [69] (www.uniprot.org (accessed on 26 January 2021)) for proteins, the PubChem CID [70] (pubchem.ncbi.nlm.nih.gov/ (accessed on 26 January 2021)) and ChEBI ID [67] (www.ebi.ac.uk/chebi/ (accessed on 26 January 2021)) for simple molecules. Furthermore, all interactors are annotated by the PMID (pubmed.ncbi.nlm.nih.gov (accessed on 6 June 2021)) of the references they are derived from. We define interactors as molecular entities that influence CFTR either through direct physical interactions or indirect regulatory interactions.

In order to provide additional information, all interactors in the map are color-coded. Blue entities represent CFTR or versions thereof (gene, mRNA, protein). All interactors that could be confirmed in at least one polarized cell line are depicted using a green color scheme, and interactors only identified in non-polarized cell lines are shown in yellow.

### 4.2. Literature Curation for the Core Map

In order to assess the current state of knowledge for CFTR maturation and activity, a literature overview was created, starting from 23 relevant main reviews of the last two decades, which can be found in Appendix A. Highly cited examples of these reviews are Riordan, 2008 [71], Lukacs and Verkman, 2012 [25] and Farinha and Canato, 2017 [72], which cover the expression, folding, maturation and function of CFTR in great detail and were published over the course of nearly one decade, thereby providing extensive descriptions of recent as well as earlier findings. From there, a preliminary consensus network of CFTR-relevant pathways and a list of resulting interaction partners was created. To validate the list of interactors, the literature database PubMed [73] was searched for experimental publications that confirmed the interaction through small-scale experiments using methods such as co-immunoprecipitation, pull-down assays, two-hybrid assays, NMR, X-ray crystallography, surface plasmon resonance, and functional (mutational) assays. A complete list of the 221 references published between 1991 and 2020, which were used for validation, can be found in Appendix A. An interactor was accepted for the model when it could be identified in at least two small-scale experiments conducted in human cells from independent references or was considered as acknowledged by the research community when described in at least two reviews from different research groups. For example, the interaction of CFTR with the gene product of *SLC9A3R1* (also known as NHRF1 or EBP50) at the plasma membrane is well documented in experimental publications as well as reviews [72,74,75,76], and it was therefore accepted for the CFTR core map. On the other hand, the serum response factor (SRF) has been reported to act as a transcription factor for CFTR by René et al. [77] but, to our knowledge, has not yet been confirmed by other research groups. Consequently, it was not included in the CFTR core map for now. This does not mean that SRF is not considered a CFTR interactor; it is merely a quality control check to ensure a high data quality and high evidence of the interactions compiled in the core map.

### 4.3. Integration of Protein–Protein Interaction Databases

To ensure that no high-confidence interactors were missed, the manually curated list of interactors was then complemented with data from small-scale experiments from existing protein–protein interaction databases (SIGNOR2.0 [78], BioGRID [79], the Human Protein Reference Database [80], String DB [81], MINT [82], InnateDB [83], APID [84] and IntAct [85]). The same literature criteria were applied. References to large-scale experiments were excluded from the database searches for the core map and later added as a second data layer in the coarse map (see Section 4.5).

### 4.4. Consideration of Cell Polarity

The proper maturation and integration of functional CFTR into the plasma membrane is highly dependent on the polarization of the cell [51]. In order to take this into account in our model, the experimental method was specified and the cell lines used were listed for each interaction. Cell lines were divided into non-polarized and polarized cells and it was specified for each interaction whether or not it was shown in at least one polarized cell line. Under experimental conditions, the question whether cells are to be characterized as polarized or non-polarized can often be a matter of debate, which is beyond the scope of this study. Therefore, polarized cells are here defined as cells with the general ability to polarize. A complete list of interactions and the respective publications with the experimental method and cell line used can be found in Appendix A. During the literature curation, interactors were successively added to the list and model, which was visualized using the diagram editor CellDesigner4.4.2 [49,50] and the cell-polarity was indicated by color. Therefore, our model rudimentarily takes into account that a range of different model systems was used to study the CFTR lifecycle.

### 4.5. Visualization of the High-Throughput Interactome as Coarse Map

In the last 15 years, many potential interactors of CFTR have been identified through high-throughput methods [45,46,47,48]. While these methods are able to generate high amounts of data, they provide less information on the nature of the interaction and the confidence of the interaction is lower than through the identification in several small-scale studies, as more false-positives may occur. In order to address this discrepancy in data quality and still provide an extensive interaction map, the wt-CFTR interactomes published by Wang et al., 2006, Pankow et al., 2015, Santos et al., 2019 and Matos et al., 2019 [45,46,47,48] were added as separate data layer from large-scale experiments. Only the wt-specific interactome was considered, and interactors unique to the F508del-variant were excluded. Missing information on the subcellular localization of the interactors reported was gathered from UniProt and the Human Protein Atlas, using the respective application programming interface, to update to the knowledge on subcellular localization in 2021. The subcellular localizations specified in the published data were combined into more general main localizations (nucleus, endoplasmic reticulum (ER), Golgi apparatus, endosome, plasma membrane, mitochondrion, cytoplasm, and extracellular space). Based on the functional categorization specified in the published data, the interactors were grouped into general functional categories (DNA replication, transcription, RNA processing, translation and folding, ER-associated degradation (ERAD), trafficking pathway, cytoskeleton and stabilization, activity and regulation, recycling, degradation, immune response, and other). The functional categorization was solely based on the information provided in the publications and not inferred from other sources. Interactors already present in the core map (46 interactors) were excluded in the coarse map to avoid redundancies between the data layers. A complete list of the interactors from the large-scale experiments, together with their subcellular localization and functional categorization, can be found in Appendix A. The list of interactors was split according to their functional categorizations and visualized in individual maps using the SBGN Activity Flow notation through the Python library libsbgnpy 0.2.2 [52] and CellDesigner4.4.2 [49,50]. 

Through this procedure, the findings from high-throughput efforts are available in the model, while the different data-quality, compared to the manually validated interaction partners, becomes visible for the user. While the high-throughput data have a lower confidence, they are also free from prior assumptions, whereas the small-scale experiments offer a high confidence but stem from restricting hypothesis-driven approaches.

### 4.6. Analysis of the Protein–Protein Interaction Network within the CFTR Core Map

To analyze the manually curated core map with regard to cross-interactions between the proteins included, a protein–protein interaction network was created using the list of genes present in the core map. Physical interactions between the manually curated interactors of CFTR were identified in GeneMania [86] using only physical interactions reported by BioGrid-small scale studies [79]. The complete list of interactions can be found in Appendix A. The network was visualized using the Python plotting library Matplotlib [87] and analyzed as a weighted, undirected graph using the Python packages NetworkX [88] and python-louvain [89].

### 4.7. Comparison of Content Provided to the Model by Small-Scale and Large-Scale Experiments 

BioInfoMiner [53] was used to analyze and compare the gene lists derived from the manually curated core map and from the coarse map containing the interactomes published by Wang et al. [45], Pankow et al. [46], Santos et al. [47] and Matos et al. [48]. The BioInfoMiner tool provides a topological analysis of semantic networks and prioritizes key systemic processes and related genes present in a set of genes. Here, analysis based on Gene Ontology [54,55] and the Reactome Pathway Database [56] were conducted. The Reactome Pathway Database represents molecular pathways that are part of human biological processes, while Gene Ontology also assigns broader biological process terms to genes and gene products. Hence, the Reactome Pathway Database mainly associates genes and gene products to rather specific terms (e.g., *N-glycan trimming in the ER and Calnexin/Calreticulin cycle*), whereas they might be assigned to more general terms in Gene Ontology (e.g., *Protein Folding*).

## Figures and Tables

**Figure 1 ijms-22-07590-f001:**
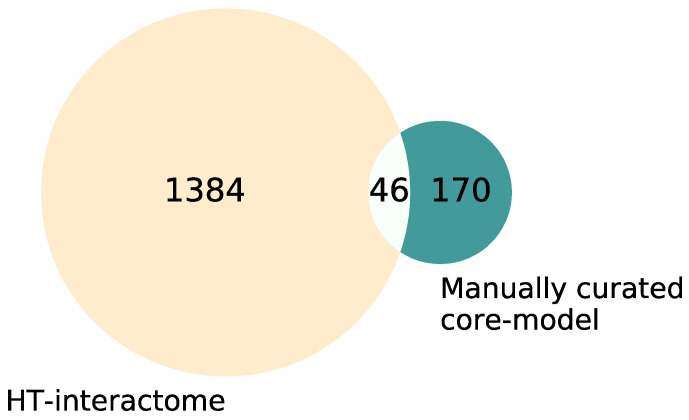
Venn diagram of the interactors in the manually curated core map vs. coarse map. The manually curated list of interactors within the core map comprises 170 interactors, the coarse map derived from high-throughput data contains 1384 interactors, and 46 interactors occur in both lists. The overlap was subtracted from the high-throughput interactome for the visualization to avoid redundancies, resulting in 1338 interactors in the coarse map.

**Figure 2 ijms-22-07590-f002:**
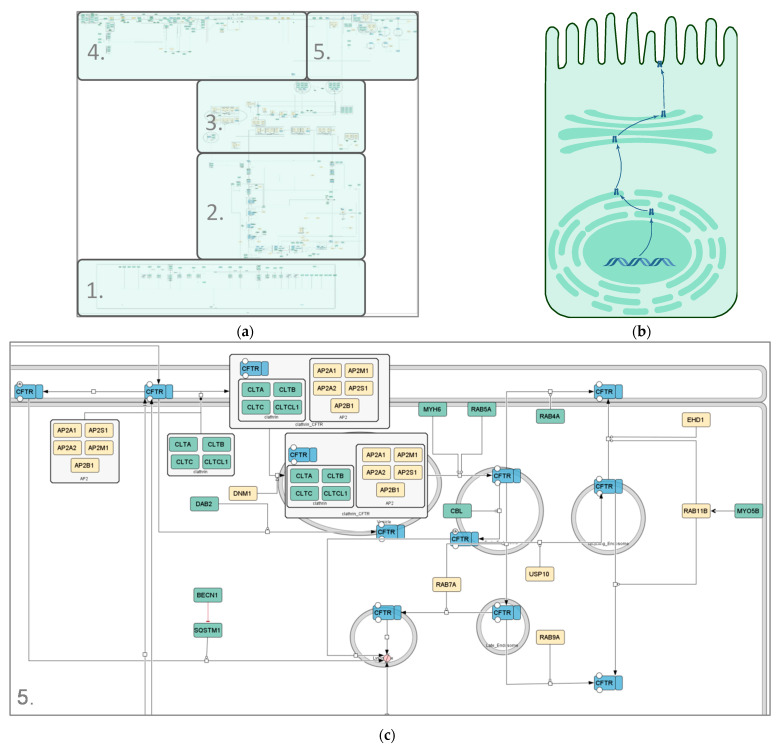
Different representations of the CFTR lifecycle. (**a**) Image of the SBGN-compliant manually curated CFTR lifecycle core map; (**b**) cartoon representation of the CFTR lifecycle in an apical epithelial cell; (**c**) zoomed-in section of the endocytosis pathway in the manually curated CFTR lifecycle core map (submap 5). Compartments are depicted in grey, CFTR in blue and interactors in differing shades of green and yellow. The green color scheme represents interactors identified in at least one polarized cell line; yellow interactors were identified in non-polarized cell lines only. State transitions, catalysis and positive influences are shown in black; negative influences and inhibitions are displayed in red. Different shapes represent different kinds of interactors. Rounded rectangles correspond to proteins, ovals and circles to small molecules and ions, respectively, rectangles correspond to genes, rhomboids to RNA molecules and chevron shapes to receptors. The map was created using CellDesigner4.4.2.

**Figure 3 ijms-22-07590-f003:**
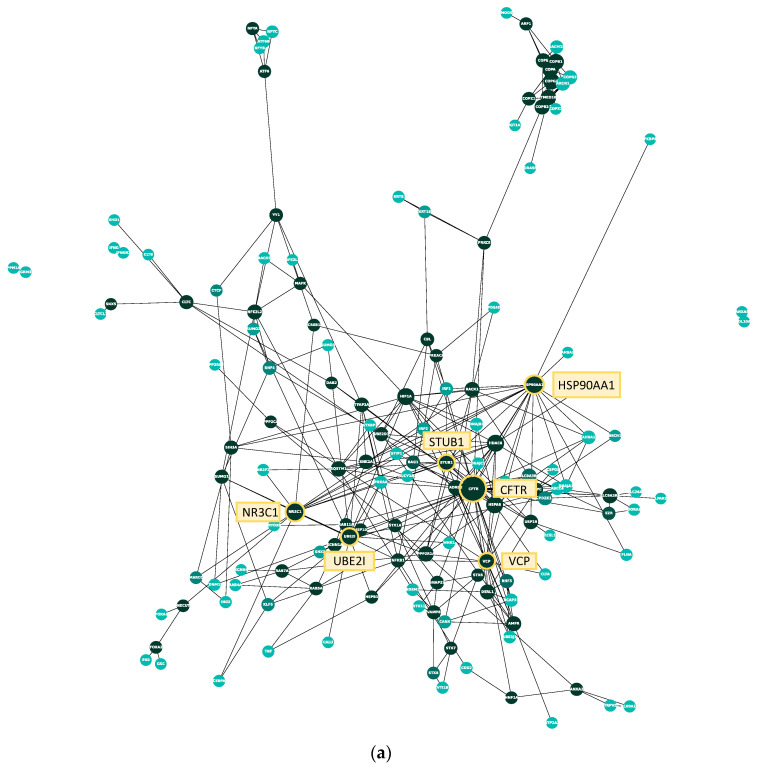
Protein–protein interaction network and degree distribution of the manual CFTR map. (**a**) Each node represents one protein, each edge between them a physical interaction shown in a small-scale study and reported on BioGrid. The larger a node, the higher its degree (i.e., the more interactions it shares with other proteins). CFTR and the five proteins with the next-highest degree are marked in yellow. The color of the protein represents its betweenness centrality, which is a measure of how important the node is to the flow of information through the CFTR Lifecycle Map. The betweenness centrality of a protein is the number of times it lies on the shortest path between two other proteins. The darker the node, the higher its betweenness centrality; (**b**) bar plot of the degree distribution of the protein–protein interaction network in A. The x-axis shows the degree of a protein; the degree is the number of other proteins a protein interacts with. The y-axis shows the number of proteins in the network with a certain degree. For example, the number of proteins that interact with only one other protein in the network (i.e., have a degree of 1) is 30, the number of proteins that interact with six other proteins is ten. The node with the highest degree (38) is CFTR.

**Figure 4 ijms-22-07590-f004:**
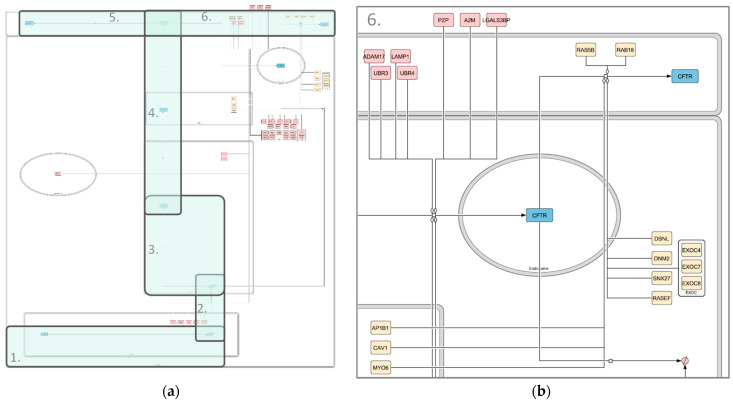
Image of the high-throughput CFTR Endocytosis, Recycling and Degradation map. (**a**) Whole map; (**b**) zoomed-in section of the map. Compartments are depicted in grey, CFTR in blue and interactors in shades of red (degradation associated) and yellow (recycling associated). State transitions and modulations are shown in black. Different shapes represent different kinds of interactors. Rounded rectangles correspond to proteins, rectangles correspond to genes and rhomboids to RNA molecules. The map was created using libsbgnpy 0.2.2 and CellDesigner4.4.2.

**Figure 5 ijms-22-07590-f005:**
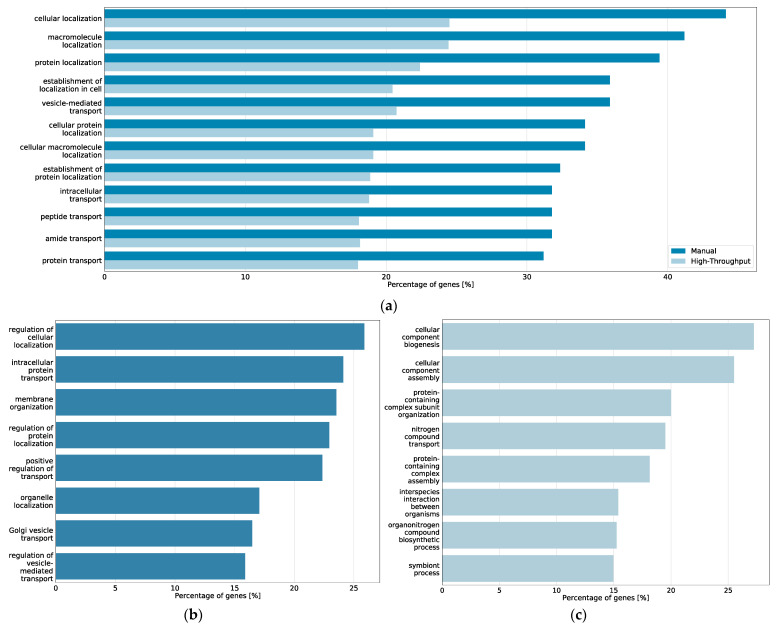
Top 20 prioritized Gene Ontology processes from BioInfoMiner analysis of the CFTR Lifecycle Map. (**a**) Bar plot of the processes prioritized among the top 20 in both the core map and the coarse map datasets and percentage of genes from the respective map associated with the processes; (**b**) bar plot of the processes prioritized among the top 20 in only the manually curated gene list of the core map and percentage of genes from the core map associated with the processes; (**c**) bar plot of the processes prioritized among the top 20 in only the high-throughput interactome of the coarse map and percentage of genes from the coarse map associated with the processes.

**Table 1 ijms-22-07590-t001:** Description of the five submaps of the CFTR core map.

Process	Localization	Molecular Entities Present in the Model	N ^1^	Proportion of the Interactors Identified in Polarized Cells
Transcription	Nucleus	Proteins	28	97%
		RNAs and gene elements	16	
		Small molecules and ions	1	
Translation, Folding and ER quality control	ER	Proteins	45	69%
		Small molecules and ions	13	
Secretory pathway	ER, Golgi apparatus, Plasma Membrane	Proteins	27	52%
		Small molecules and ions	8	
Activity and Regulation	Plasma Membrane	Proteins	44	82%
		Small molecules and ions	20	
Endocytosis, Recycling and Degradation	Plasma Membrane, Endosomes, Lysosomes	Proteins	23	74%

^1^ Number of different molecular entities present in the respective submap.

**Table 2 ijms-22-07590-t002:** Description of the five submaps of the CFTR high-throughput model.

Process	Localization	Functional Category	N ^1^
Transcription	Nucleus	DNA Replication	7
		Transcription	10
RNA Processing	Nucleus–Cytoplasm		36
Translation, Folding and ER quality control	ER	Folding	57
		ER-associated degradation	1
		both	3
		unspecified	653
Secretory pathway	ER, Golgi apparatus, Plasma Membrane		22
Activity and Regulation	Plasma Membrane	Activity	38
		unspecified	145
Endocytosis, Recycling and Degradation	Plasma Membrane, Endosomes, Lysosomes	Recycling	12
		Degradation	32
Cytoskeleton	Cytoplasm, Plasma Membrane		62
Immunity			10
Other/Unknown			250

^1^ Number of interactors present in the respective submap.

## Data Availability

The data presented in this study are available in the Appendix A. Future versions of the CFTR Lifecycle map will be made available here https://candactcftr.ams.med.uni-goettingen.de/SystemsBiology/ (accessed on 26 January 2021).

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
