# Peer review of "CFTR Lifecycle Map—A Systems Medicine Model of CFTR Maturation to Predict Possible Active Compound Combinations"

_ijms, 2021, doi:10.3390/ijms22147590_

Round 1

Reviewer 1 Report

This paper provides a very useful tool for CF research: it summarizes all the main findings in the area. I think that some sentences should be re-written to make them clearer; for example, from the sentence in Line 40-42 it is not clear what the ABC proteins are supposed to do with ATP. 

I also suggest that the authors add a schematic figure in the introduction to better explain the the structure  of CFTR and the classification of the 6 classes of the CF-causing mutations. 

Moreover, I would reccomend to clarify the aim of this paper: why they decided to create this tool? Why it should be useful? 

The explanation of PD in line 444-452 should be mover elsewhere (maybe in the introduction?). 

In the end, the authors should better explain the reason why they have chosen the papers cited in line 474 as the most relevant. 

Author Response

This paper provides a very useful tool for CF research: it summarizes all the main findings in the area. I think that some sentences should be re-written to make them clearer; for example, from the sentence in Line 40-42 it is not clear what the ABC proteins are supposed to do with ATP.

We thank the reviewer for the kind assessment of our tool and recognising its usefulness.

We have rewritten the text passage in the introduction describing ABC proteins, their dependence on ATP and the unique features of CFTR. Specifically, we now explain the difference between ABC transporters and the ion channel CFTR with respect to how ATP is utilized.

I also suggest that the authors add a schematic figure in the introduction to better explain the the structure  of CFTR and the classification of the 6 classes of the CF-causing mutations.

We added references to some in-depth reviews on the structure of CFTR and classification of CF-causing mutations to the relevant text passages. However, due to the highly specialized nature of this special issue on “Cystic Fibrosis and CFTR Interactions”, we refrained from adding figures on the matter. The reviews cited provide excellent figures and explanation and we believe more in-depth descriptions of CFTR structure and mutation classifications would be superfluous and beyond the scope of this manuscript.

Moreover, I would reccomend to clarify the aim of this paper: why they decided to create this tool? Why it should be useful?

We added another paragraph to the end of the introduction which emphasizes the aims and relevance of this manuscript. We stated our main motivation to create this tool and possible applications for the CF-research community to clarify its usefulness. Specifically, we now emphasize that CFTR Lifecycle map provides a tool to structure and exploit existing knowledge and data, as well as develop hypothesis regarding synergistic drug targets and novel therapeutics.

The explanation of PD in line 444-452 should be mover elsewhere (maybe in the introduction?).

We thank the reviewer for the suggestion and added a paragraph explaining Process Description (PD) to the introduction. The paragraph introduces the PD language as sub-language of the SBGN notation and describes its specificities. We also give some examples for the manner in which molecular interactions and reactions are depicted in SBGN-PD language.

In the end, the authors should better explain the reason why they have chosen the papers cited in line 474 as the most relevant.

We rephrased and added to the paragraph on the literature review with a more detailed description of our approach and the publications that were used for our initial literature review. We explicitly mention the list of 23 reviews from the last two decades, which can be found in the supplementary and covers different aspects of the molecular biology of CFTR.  We now also explain the high relevance of the three papers cited which were chosen as example. They are highly cited reviews in the field and were published over the course of nearly a decade, thereby covering recent as well as earlier findings.

Reviewer 2 Report

The manuscript by Vinhoven and colleagues aim to construct a CFTR related network, by gathering relevant publications, in order to provide and potentially predict active compound combinations. Overall it is a well written article, with a clear message, mainly aiming readers in this field. It is suitable for publication at International Journal of Molecular Sciences, upon addressing the below comments.

Major points

  • Fig 5 creates some confusion. The % of genes in for example Fig 5a corresponds to all the genes responsible for example for the cellular organization, or the % corresponds to how many of CFTR lifecycle map genes are responsible for the cellular organization? Maybe the authors could consider providing also an enrichment analysis?
  • The authors should consider providing 1-2 line of explanations for systems biology related terms which are not usually used by non-experts.

Minor points

  • “The high-throughput data layer encompasses 1384 unique interactors from two publications by Wang et al., 2006, Pankow et al., 2015, Santos et al., 2019 and Matos et al., 2019 [42–45].” The authors state two publications but cite five.

Author Response

The manuscript by Vinhoven and colleagues aim to construct a CFTR related network, by gathering relevant publications, in order to provide and potentially predict active compound combinations. Overall it is a well written article, with a clear message, mainly aiming readers in this field. It is suitable for publication at International Journal of Molecular Sciences, upon addressing the below comments.

We thank the reviewer for this kind assessment of our manuscript.

Major points
Fig 5 creates some confusion. The % of genes in for example Fig 5a corresponds to all the genes responsible for example for the cellular organization, or the % corresponds to how many of CFTR lifecycle map genes are responsible for the cellular organization? Maybe the authors could consider providing also an enrichment analysis?

We thank the reviewer for pointing out the confusion and agree that the annotation of Fig 5 is ambiguous. We rewrote it accordingly, making sure it clearly states that the percentage corresponds to the proportion of genes in the respective layer of the CFTR lifecycle map that is associated to the respective biological process. The figure annotation should now be unambiguous and better understandable.

The authors should consider providing 1-2 line of explanations for systems biology related terms which are not usually used by non-experts.

In order to make the systems biology parts of the manuscript easier to understand for non-experts, we have created a glossary with more detailed descriptions of the relevant terms from the fields of systems biology and bioinformatics to the supplement. The glossary is referred to in the introduction when first introducing systems biology related terms and was assessed by a non-expert in systems biology in order to include all terms in question.  We believe this is a good way to explain all terms and give further information for the interested reader without impeding with the reading flow.

Minor points
“The high-throughput data layer encompasses 1384 unique interactors from two publications by Wang et al., 2006, Pankow et al., 2015, Santos et al., 2019 and Matos et al., 2019 [42–45].” The authors state two publications but cite five.

We thank the reviewer for pointing out the typo and corrected it.